# Structural organization of a major neuronal G protein regulator, the RGS7-Gβ5-R7BP complex

Dipak N Patil[1], Erumbi S Rangarajan[2], Scott J Novick[3], Bruce D Pascal[3], Douglas J Kojetin[2], Patrick R Griffin[2,3], Tina Izard[2]*, Kirill A Martemyanov[1]*

[1]Department of Neuroscience, The Scripps Research Institute, Jupiter, United States; [2]Department of Integrative Structural and Computational Biology, The Scripps Research Institute, Jupiter, United States; [3]Department of Molecular Medicine, The Scripps Research Institute, Jupiter, United States

**Abstract** Signaling by the G-protein-coupled receptors (GPCRs) plays fundamental role in a vast number of essential physiological functions. Precise control of GPCR signaling requires action of regulators of G protein signaling (RGS) proteins that deactivate heterotrimeric G proteins. RGS proteins are elaborately regulated and comprise multiple domains and subunits, yet structural organization of these assemblies is poorly understood. Here, we report a crystal structure and dynamics analyses of the multisubunit complex of RGS7, a major regulator of neuronal signaling with key roles in controlling a number of drug target GPCRs and links to neuropsychiatric disease, metabolism, and cancer. The crystal structure in combination with molecular dynamics and mass spectrometry analyses reveals unique organizational features of the complex and long-range conformational changes imposed by its constituent subunits during allosteric modulation. Notably, several intermolecular interfaces in the complex work in synergy to provide coordinated modulation of this key GPCR regulator.

**\*For correspondence:**
cmorrow@scripps.edu (TI);
kirill@scripps.edu (KAM)

**Competing interests:** The authors declare that no competing interests exist.

## Introduction

G-protein-coupled receptors (GPCRs) constitute the largest family of cell surface receptors that endow cells with an ability to detect and respond to a vast array of extracellular stimuli including neurotransmitters, hormones, ions, and light (*Shoichet and Kobilka, 2012*). GPCR signaling systems are involved in virtually all essential physiological processes and their dysregulation is firmly linked to a number of human diseases (*Gutierrez, 2018*; *Insel et al., 2007*; *Iyinikkel and Murray, 2018*). Accordingly, an estimated one-third of all FDA approved and marketed drugs target GPCRs for therapeutic benefits (*Hauser et al., 2017*). Therefore, understanding how GPCRs transmit their signal and how this process is regulated is of paramount importance.

Ligand-bound GPCRs transmit signals by catalyzing the exchange of GDP for GTP on the heterotrimeric G proteins thereby releasing active GαGTP and free Gβγ subunits, which in turn engage a wide range of the intracellular effectors to produce a cellular response (*Pierce et al., 2002*). A wealth of accumulated evidence indicates that the key role in regulating this process belongs to a conserved family of regulators of G-protein signaling (RGS) proteins (*Hollinger and Hepler, 2002*; *Ross and Wilkie, 2000*; *Kimple et al., 2011*). RGS proteins act as GTPase-activating proteins (GAPs) by interacting directly with activated Gα and accelerating GTP hydrolysis thereby promoting G-protein inactivation upon reassembly of the inactive Gαβγ heterotrimer. The action of the RGS proteins is essential in providing control over both the extent and duration of signaling (*Hollinger and Hepler, 2002*; *Ross and Wilkie, 2000*) and deficits in this control results in severe dysregulation of

GPCR signaling causing a number of human pathophysiological conditions (*Woodard et al., 2015*; *Salaga et al., 2016*; *Druey, 2017*; *Sjögren, 2017*).

More than 30 RGS proteins have been identified and grouped into six subfamilies (*Zheng et al., 1999*). Among them, the R7 family (RGS6, RGS7, RGS9, and RGS11) stands out for its evolutionary conservation in all animals from worm to man and crucial roles in multiple processes and organ systems including nervous and cardiovascular system function, vision, movement control, and cellular proliferation with ever growing causal connection to many diseases from blindness to cancer (*Ahlers et al., 2016*; *Qutob et al., 2018*; *Yang et al., 2013*; *Anderson et al., 2009b*). A unique hallmark of R7 RGS proteins is their modular architecture encompassing several domains and subunits. The catalytic RGS domain at the C-terminus is the site for the interaction with the Gα subunits (*Hooks et al., 2003*). The central GGL domain forms an obligatory heterodimer with type 5 G-protein β-subunit (Gβ5) which stabilizes the complex and docks it onto effectors (*Snow et al., 1998*; *Xie et al., 2010*). In addition, the molecule contains the N-terminal interwoven DEP (Dishevelled, Egl10, Pleckstrin) and DHEX (DEP helical extension) domains, a site for major allosteric regulation of the complex via association with GPCRs (*Orlandi et al., 2012*; *Sandiford and Slepak, 2009*; *Kovoor et al., 2005*) and small SNARE-like membrane proteins: the R7-binding protein (R7BP) and/or RGS9-anchor protein (R9AP) (*Hu and Wensel, 2002*; *Drenan et al., 2005*; *Song et al., 2006*; *Martemyanov et al., 2005*). This complexity of organization is thought to reflect regulatory precision and flexibility adapting R7 RGS proteins to universally meet diverse needs of multiple GPCR pathways in changing molecular landscape of different cells. Yet, our understanding of the structural basis underlying functional diversity is limited. The only structure reported to date is that of the RGS9-Gβ5 dimer, a specialized R7 RGS member with expression limited to striatal neurons and photoreceptors. Furthermore, no structural information exists for the regulation of RGS proteins by R7BP/R9AP subunits.

RGS7 is a prototypic member of the family, broadly expressed in the nervous system and periphery and involved in regulation of multiple GPCR systems with key roles in physiology. A few documented examples include regulation of mGluR in retina bipolar (*Cao et al., 2012*), GABA(B) in hippocampal CA1 (*Fajardo-Serrano et al., 2013*; *Ostrovskaya et al., 2014*), μ-opioid in striatal neurons (*Sutton et al., 2016*; *Anderson et al., 2010*; *Masuho et al., 2013*), and M3 muscarinic receptors in the pancreatic β-cells (*Wang et al., 2017*). Given these roles, perhaps not surprisingly, the RGS7 complex has been found to play a critical role in vision, learning and memory, drug addiction, insulin production, and cancer progression (*Qutob et al., 2018*; *Sutton et al., 2016*; *Wang et al., 2017*). Furthermore, there is a growing appreciation that the RGS7 complex can be exploited as a target for drug development with the hopes of increasing efficacy, selectivity, and the side effect profile of existing GPCR medications (*Muntean et al., 2018*). Thus, so far the lack of structural information on organization and regulation of the RGS7 complex has been a major bottleneck in understanding basic biology of GPCR signaling regulation and rational design of small molecule modulators.

Here, we report the high-resolution crystal structure of the RGS7-Gβ5 dimer and the structural basis for its association with R7BP revealed by hydrogen-deuterium exchange as detected by mass spectrometry (HDX-MS) analyses. In combination with molecular dynamics simulations, our data reveal unique structural features of the RGS7 complex that make it suitable for the regulation of diverse GPCR target pathways. These include the long-range rearrangement in the complex induced by R7BP binding and of the complex providing structural basis for its allosteric regulation. Together our findings reconcile a wealth of biochemical observations pertaining to the function of RGS7 thereby providing crucial insights into the biology of this essential GPCR signaling regulator.

## Results

### Overall structural organization of the RGS7-Gβ5 heterodimer

To gain structural insights into the specificity of the RGS7-Gβ5 interaction, we solved the 2.13 Å crystal structure of the RGS7-Gβ5 dimer complex which required buffer optimization using thermofluor (*Figure 1*, *Tables 1* and *2*). The structure contains all domains common to the R7 RGS family members: DEP-DHEX, GGL and RGS (*Figure 1*). The core of the molecule is formed by the Gβ5 subunit, which has a typical β-propeller fold commonly found in domains with WD40 repeats. Notably, Gβ5 is

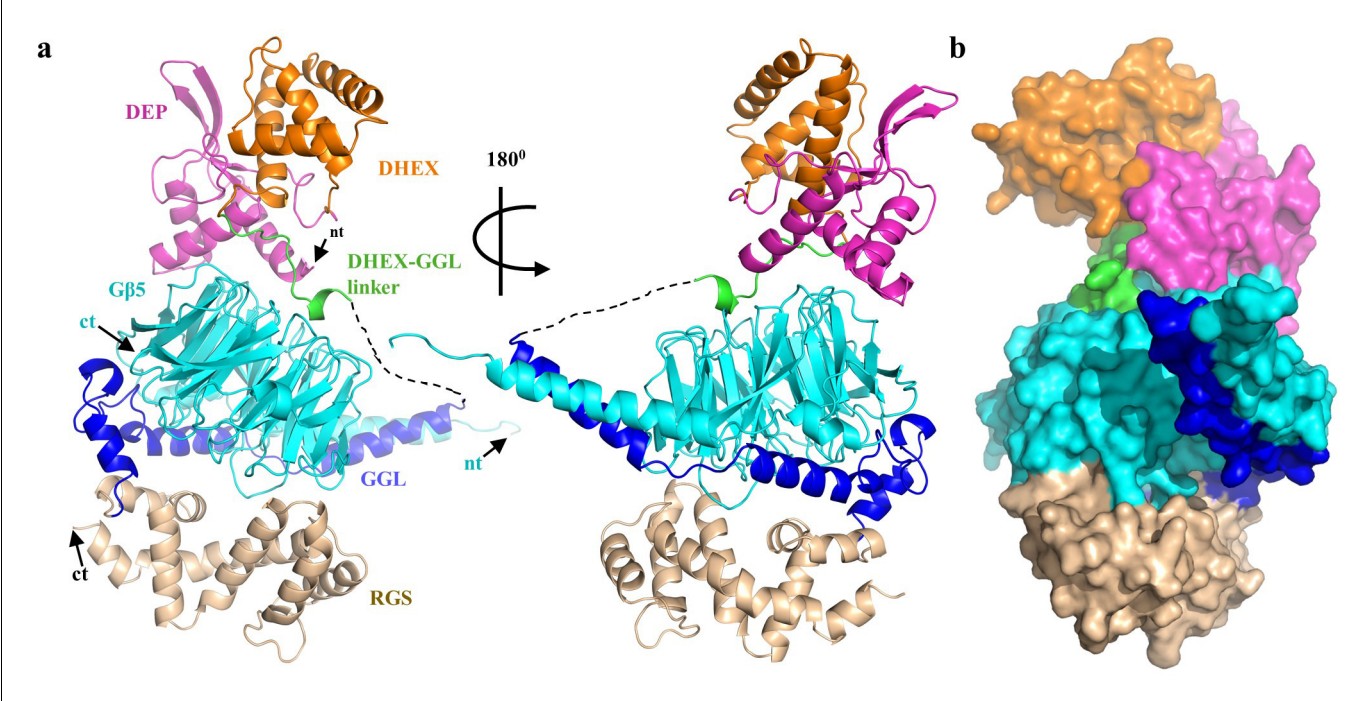

**Figure 1.** Overall structure of RGS7-Gβ5. (**a**) Cartoon representation of the RGS7-Gβ5 complex with the RGS7 domains colored individually (DEP, pink; DHEX, orange; GGL, blue; RGS, sand). Gβ5 is shown in cyan and is sandwiched between the RGS7 domains. The missing part of the DHEX-GGL linker is shown as a dotted line. (**b**) Overall surface representation of the RGS7-Gβ5 complex colored as in panel A.

The online version of this article includes the following figure supplement(s) for figure 1:

**Figure supplement 1.** Contacts between residues of the RGS7 domains and Gβ5.

engulfed by all RGS7 domains, making extensive contacts at four main interfaces: the DEP domain, the DHEX-GGL linker, the Gγ-like (GGL) domain, and the RGS domain (*Figure 1—figure supplement 1*). Two major distinctive features of RGS7-Gβ5 include a unique arrangement of the DEP-DHEX and GGL-Gβ5 modules, discussed in detail below. The total buried solvent-accessible surface area of Gβ5 is 29% by RGS7 domains and involves 108 out of 353 Gβ5 amino acids. Electron density is visible for residues 19–218 and 252–450 of chain A as well as 19–218 and 252–450 of chain B of RGS7 and for residues 2–353 of chain C and 4–353 of chain D of Gβ5. Overall, the structure shows extensive coordination between multiple RGS structural features integrated around the central 'scaffolding' element provided by the Gβ subunit.

## Distinct conformational arrangement of the DEP-DHEX module

The DEP domain forms the N-terminus of RGS7 protein. DEP domains are present in a number of signaling proteins and play diverse functions with the common role of targeting signaling proteins to specific membranous subcellular compartments (*Kovoor et al., 2005*; *Consonni et al., 2014*; *Paclíková et al., 2017*; *Chen and Hamm, 2006*; *Martemyanov et al., 2003*; *Ballon et al., 2006*; *Civera et al., 2005*). Despite limited amino acid sequence identity, the DEP domain of RGS7 shows high structural similarity with the DEP domains of other proteins with known structures featuring a characteristic α/β fold (*Figure 2—figure supplement 1a*). It features a three helix bundle covered by an antiparallel β-sheet and a flexible loop (the Dα1Dα2 loop). In several available atomic structures of DEP domains, this flexible loop forms a protruding β-hairpin arm. Superposition of DEP domains of RGS7, RGS9, DVL, EPAC, and pleckstrin shows considerable structural diversity in the organization of this loop known for its involvement in molecular interactions (*Figure 2a*). Interestingly, in RGS7, the Dα1Dα2 and Eα1Eα2 loops adopt a unique conformation in both molecules within asymmetric unit documenting yet another alternative organization and intrinsic flexibility of this region (*Figure 2—figure supplement 1b*).

**Table 1.** X-ray data reduction statistics for the RGS7-Gβ5 dimer.

| Isotropic scaling | |
|---|---|
| Space group | P 2₁ |
| Unit cell dimensions | |
| a, b, c, β | 68.32 Å, 162.76 Å, 95.29 Å, 98.36˚ |
| Resolution (last shell) | 94.28 Å - 2.81 Å (2.86 Å - 2.81 Å) |
| Total measurements | 191,664 |
| No. of unique reflections (last shell) | 50,669 (2,501) |
| Wavelength | 1.0 Å |
| $R$p.i.m.* (last shell) | 0.036 (0.431) |
| I/σ(I) (last shell) | 15.1 (2.3) |
| Completeness (last shell) | 0.997 (1.0) |
| Multiplicity (last shell) | 3.8 (3.9) |
| CC1/2 † (last shell) | 0.999 (0.868) |
| Anisotropic scaling | |
| Resolution (last shell) | 4.28 Å - 2.13 Å (2.48 Å - 2.13 Å) |
| Total measurements | 20,4124 |
| No. of unique reflections (last shell) | 54,345 (2,718) |
| Wavelength | 1.0 Å |
| $R$p.i.m.* (last shell) | 0.038 (0.5) |
| I/σ(I) (last shell) | 14.6 (2.1) |
| Completeness (spherical) (last shell) | 0.47 (0.064) |
| Completeness (ellipsoidal) (last shell) | 0.92 (0.63) |
| Multiplicity (last shell) | 3.8 (3.7) |
| CC1/2 † (last shell) | 0.999 (0.546) |

*Precision-indicating merging $R$ factor (**Weiss and Hilgenfeld, 1997**; **Weiss, 2001**).
†Pearson correlation coefficient calculated between the average intensities of each random half of measurements of unique reflections (**Karplus and Diederichs, 2012**).

In the structure, the DEP domain is tightly integrated with the DHEX domain, a unique structural module present only in R7 RGS proteins. Overall, it resembles the four-helix bundle organization reported for RGS9 (**Cheever et al., 2008**). In both DHEX domains, three antiparallel α-helices wrap around one α-helix at the center forming the characteristic helix wrap. The fourth α-helix of DHEX interfaces with the DEP domain via contacts with the β-sheet and flexible loops of the DEP domain.

Despite the overall similarity, the organization of the DEP-DHEX module in RGS7 is strikingly different from that seen in RGS9 (**Figure 2b**). The most distinctive feature is the different orientation of the DHEX domain relative to DEP. In RGS7, the DHEX domain is lifted upward and away from Gβ5 thereby creating a wider groove between DHEX and Gβ5 when compared to RGS9. Moreover, the organization of all four α-helices is distinct. The DHEX domain of RGS7 has shorter α-helices and longer connecting loops compared to RGS9. Most prominently, this includes much longer Eα1Eα2 and DEP-DHEX loops that adopt different conformations (**Figure 2b**). Together, this distinct organization of the Eα1Eα2 and Dα1Dα2 loops positions them close to the invariant Eα3Eα4 loop to assemble a strong cluster of basic residues on the surface of the DEP-DHEX interface (**Figure 2c**). Such a structural bow is absent in RGS9 where basic residues are scattered without forming organized basic patch (**Figure 2c**). Thus, both relative orientation of DEP and DHEX domains and organization of individual constituent structural elements contribute to distinct features of the RGS7.

**Table 2.** Crystallographic refinement statistics for the RGS7-Gβ5 dimer.

| Resolution Å (last shell) | 37.54 Å - 2.13 Å (2.16 Å - 2.13 Å) |
| --- | --- |
| No. of reflections (working set) | 54,330 |
| No. of reflections (test set) | 2610 |
| *R*-factor (last shell) | 0.174 (0.296) |
| *R*-free (last shell) | 0.216 (0.451) |
| No. of polypeptide chains | 4 |
| No. of non-hydrogen atoms: | |
| Protein | 12,051 |
| solvent | 480 |
| B-factor (Å *Gutierrez, 2018*) | |
| Protein | 55.7 |
| Solvent | 48.21 |
| Root mean square deviations (*r.m.s.d.*) | |
| Bond lengths (Å) | 0.002 |
| Bond angles (°) | 0.488 |
| Ramachandran plot | |
| favored (%) | 96.25 |
| allowed (%) | 3.75 |
| outliers (%) | 0.0 |

## Unique conformational state and interactions of the central scaffolding subunit Gβ5

Gβ5 contains identical seven bladed β-propellers (WD40 domains), as seen for the canonical Gβ subunit, Gβ1. Comparison of the RGS7 and RGS9-containing dimers reveals that the same Gβ5 protein adopts different conformations indicating flexibility and that the conformation of Gβ5 could be influenced by the identity of the associated RGS protein (*Figure 3a*). One prominent difference includes the organization of the N-terminal α-helix. In RGS9, it displays the same conformation as observed in Gβ1 but in RGS7 it protrudes outward from the β-barrel core. Another difference lies in the organization of the S5β1β2 and S3β3β4 loops which are oriented in a distinct fashion (*Figure 3a*). Interestingly, these two loops have low sequence conservation with other Gβ subunits each containing two amino acids insertions (*Figure 3—figure supplement 1*). Such unique organization likely underlies specialization in interactions with the RGS proteins, and possibly other partners, as discussed in detail below.

Notably, Gβ5 forms extensive contact interfaces with all RGS7 modules, playing a critical role in their spatial arrangement. The top surface of Gβ5 contacts two sites in RGS7: the DHEX-GGL linker and the DEP domain (*Figure 3b*). The DHEX-GGL linker interacts with the 'hotspot' region in Gβ5, which in canonical Gβ subunits is engaged in interactions with various effector molecules. In fact, there is a significant overlap between the hotspot surface in Gβ1 and the corresponding residues in Gβ5 involved in binding to the DHEX-GGL linker. Out of the 12 hotspot residues that engage in shared contacts with Gα, GRK2, and phosducin in Gβ1, seven in Gβ5 engage in interactions with the amino acids of the DHEX-GGL linker (*Figure 3c*; *Figure 3—figure supplements 1* and *2*). However, only a single residue in the Gβ5 'hotspot' overlaps with the surface that Gβ1γ2 uses to interact with GIRK (W107 in Gβ5, W99 in Gβ1), which binds to both Gβ1 and Gβ5 complexes, suggesting a unique organization of the effector binding interfaces in Gβ5 relative to canonical Gβ subunits (*Figure 3—figure supplement 3*).

The top surface of Gβ5 is also contacted by three α-helices of the DEP domain to form an extensive network of mostly hydrophobic and van der Waals interactions (*Figure 3—figure supplement 4a*). In contrast to interactions with the DHEX-GGL linker, the residues in Gβ5 in contact with the DEP domain have poor sequence conservation with other Gβ subunits and are generally outside of the 'hotspot' region (*Figure 3—figure supplement 1*) thus creating a unique interaction interface.

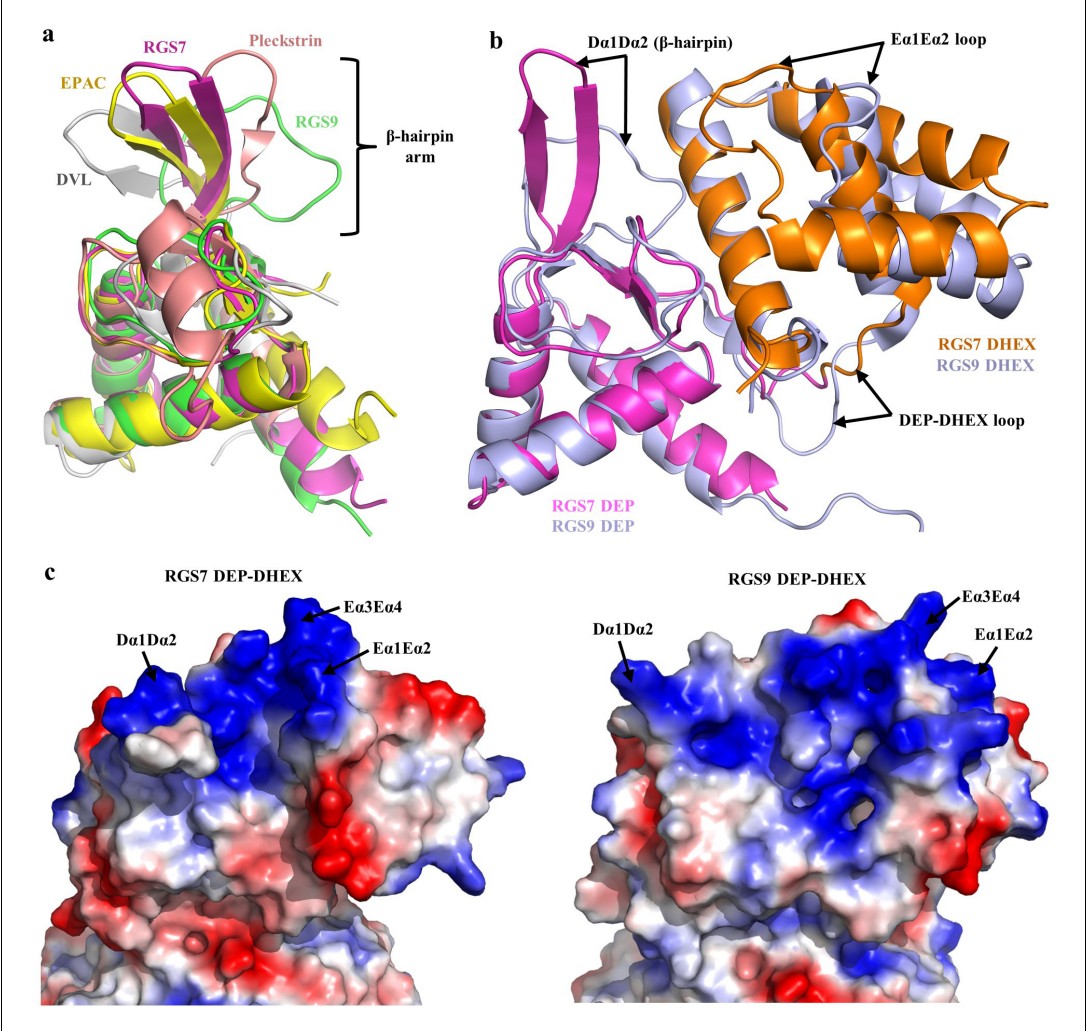

**Figure 2.** Architecture of the DEP-DHEX module. (**a**) Superposition of the DEP domains of RGS7 (pink), RGS9 (green; PDB entry 2pbi), EPAC (yellow; PDB entry 2byv), pleckstrin (brown; PDB entry 1w4m), and Dvl (grey; PDB entry 1fsh) reveals diversity in the conformational organization of the β-hairpin arm. (**b**) Comparison of the DEP-DHEX domains of RGS7 and RGS9 shows differences in several loops and in the orientation of the DHEX domain. In RGS7, the DHEX domain is lifted upward thus generating a wide space between the domains. (**c**) Evaluation of the electrostatic surface potentials of the DEP-DHEX domains of RGS7 (left) and RGS9 (right). A strong basic patch (blue) is formed by three loops in the RGS7 structure but differences in the organization of these loops result in a more scattered distribution of basic charges in the RGS9 structure.

The online version of this article includes the following figure supplement(s) for figure 2:

**Figure supplement 1.** Homology of the DEP domains and loop flexibility in the DEP-DHEX domain of RGS7.

**Figure supplement 2.** Comparison of the Dvl2 DEP domain with the Dvl2 DEP domain-μ2 complex.

Several features also distinguish the organization of the interface that the top surface of Gβ5 forms with RGS7 compared to previously observed analogous interactions with RGS9. In RGS7, the DHEX-GGL linker forms a $3_{10}$-helix embedded into the β-propeller ring at the hotspot region of Gβ5 (*Figure 3b*). In particular, three residues (D213, K215, and K216) insert deep into the ring, forming extensive contacts with surrounding residues of Gβ5 (*Figure 1—figure supplement 1* and *Figure 3— figure supplement 4b*). Such a structural feature is absent in RGS9, where the contacts are shallow. The interaction interfaces between Gβ5 and the DEP domains of RGS7 and RGS9 are also markedly different. In particular, two residues (K26 and R33) of the Dα1 α-helix of the RGS7 DEP domain form direct hydrogen bond and electrostatic interactions with D260 and E280 in Gβ5 (*Figure 3d*). These interactions are not present in the RGS9-Gβ5 complex. Instead, Gβ5 engages in electrostatic interactions with the Dα2 α-helix of RGS9 via a single ion pair of R62 in RGS9 and E280 in Gβ5 (*Figure 3d*).

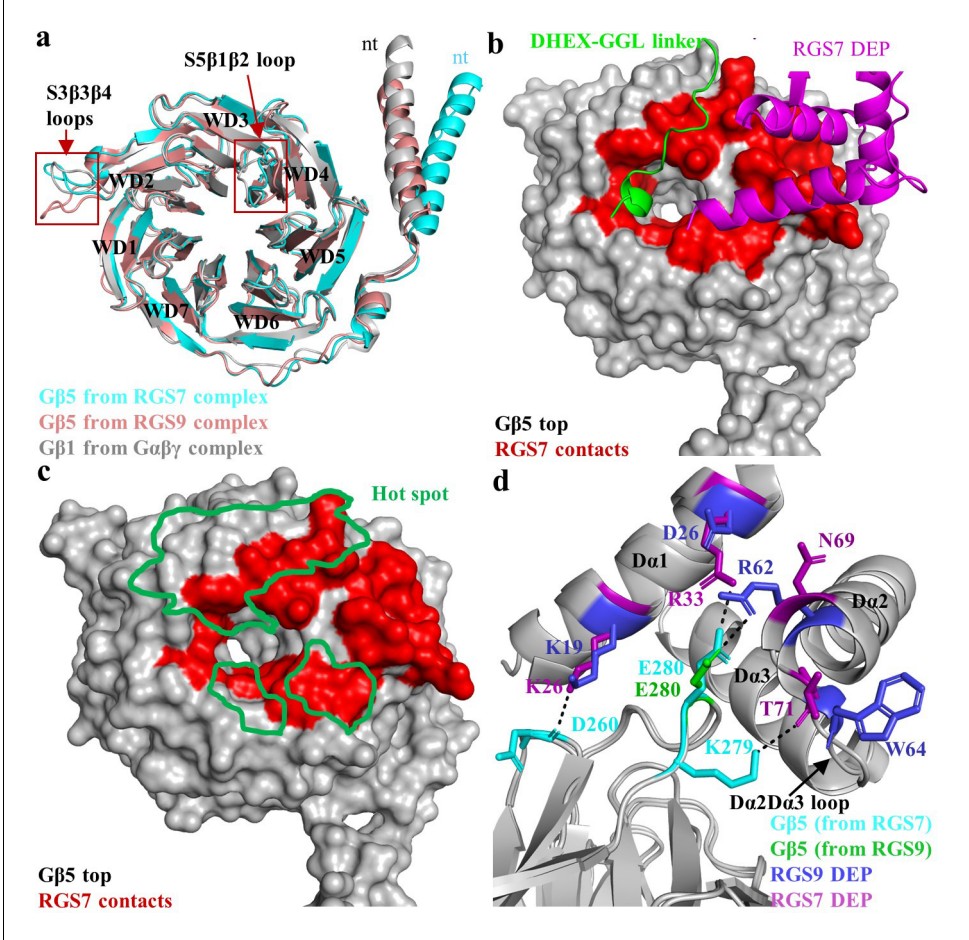

**Figure 3.** Interaction of RGS7 with Gβ5. (a) Cartoon representation of the superposed Gβ5 structures from RGS7 with RGS9 and Gβ1. The differences between the loop regions are highlighted in red boxes. The N-terminal α-helix of Gβ5 from RGS7 also shows variation compared to Gβ5 from the RGS9 complex and Gβ1 from the Gαβγ complex. (b) Surface representation of Gβ5 top face (grey) highlighting the RGS7 contact residues (red). Two contact interfaces that are formed by the DHEX-GGL linker and the DEP domain are shown as a carton in green and pink, respectively. (c) The hotspot region (green) mapped onto the Gβ5 surface based on contacts formed by known Gβ1 binding partners. RGS7 (red) overlaps with the hotspot footprint. The DEP domain forms a unique contact surface on the top face of Gβ5. (d) Distinct interacting residues of the DEP domain of RGS7 (pink) that forms direct contacts with Gβ5 (cyan) are shown along with equivalent residues of RGS9. A unique electrostatic interaction is formed by Dα1 residue R33 of the RGS7 DEP domain with E280. Its organizational equivalent in RGS9 is formed by interactions with R62 from α-helix Dα2. The Dα2-Dα3 loop of the DEP domain that is involved in interactions with Gβ5 has a different organization in RGS7 and RGS9 and is indicated by an arrow.

The online version of this article includes the following figure supplement(s) for figure 3:

**Figure supplement 1.** Gβ subunit sequence alignments and contact residues with effector molecules.

**Figure supplement 2.** Comparison of binding interfaces of various effector molecules of the Gβ subunits with RGS7-Gβ5 by mapping the overlap onto the Gβ5 surface.

**Figure supplement 3.** Comparison of the binding interfaces of RGS7 with GIRK2 of the Gβ subunits.

**Figure supplement 4.** Interaction of RGS7 with the Gβ5 subunit.

The organization of the Dα2-Dα3 loop is also markedly different in RGS7. It is longer than in RGS9 (*Figure 3—figure supplement 4c*) resulting in an additional hydrogen bond contact between T71 in RGS7 with K279 in Gβ5 (*Figure 3d* and *Figure 1—figure supplement 1*). Together, these structural features result in more extensive contacts of Gβ5 with the DEP domain of RGS7 compared to RGS9.

On the opposite side from the hotspot region, the 'bottom' of the Gβ5 contacts the GGL and RGS domains of RGS7 (*Figure 4a*). The GGL domain forms a central part of RGS7 and

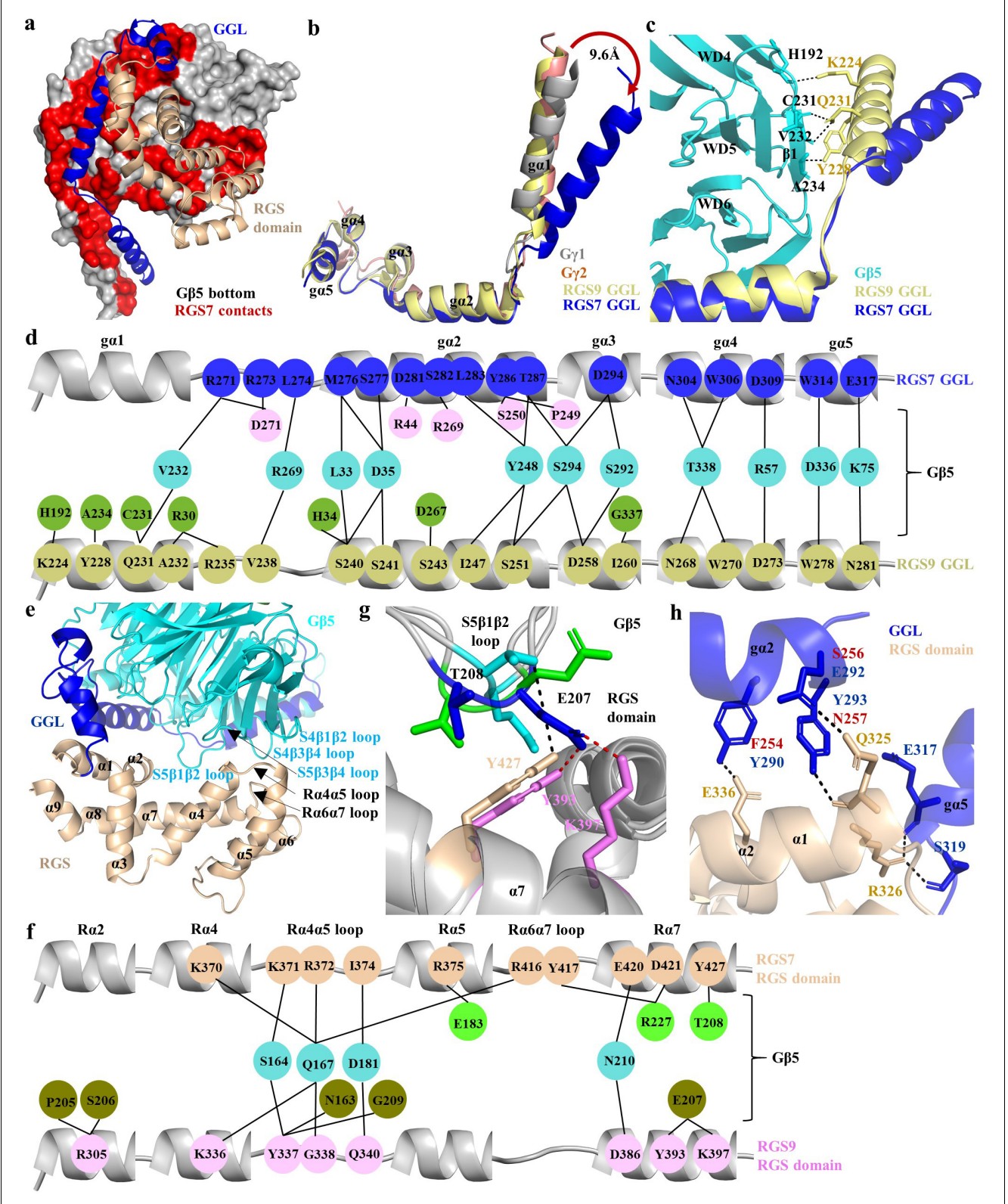

**Figure 4.** Unique mode of the Gβ5 interaction with the GGL domain of RGS7. (a) Surface representation of Gβ5 (bottom face; grey) with RGS7 contact residues (red) formed by GGL (glue) and the RGS domain (sand) is shown. (b) Comparison of the GGL domains in RGS7 and RGS9 with Gγ1 and Gγ2. (c) The Gα1 α-helix of RGS7 is located 9.6 Å away from its position in RGS9, Gγ1, and Gγ2 diminishing the contacts of the Gα1 α-helix with Gβ5. (d) Schematic view of detailed comparison of the direct contact residues of GGL of RGS7 and RGS9 with Gβ5 residues. Most of the contact residues of

*Figure 4 continued on next page*

*Figure 4 continued*

Gβ5 are conserved (cyan); however, the RGS7 Gα1 α-helix does not form any contacts. The Gα1-Gα2 and Gα2 α-helices contribute to forming additional contacts (pink) that consist of two electrostatic interactions which are unique to RGS7. The RGS9-specific residues of Gβ5 are shown in green color. (e) Cartoon representation of the contact interface of RGS domain (orange) with Gβ5 (cyan) and GGL (blue) are shown. The α-helices of the RGS domain are numbered as α1-α9. Loops are indicated by black arrows. (f) Schematic view of the comparison of the direct contact residues of the RGS domains of RGS7 and RGS9 with Gβ5. Common contact residues of Gβ5 are shown in cyan, RGS-specific residues in green, and RGS9 in olive. (g) The S5β1β2 loop of Gβ5 (cyan and green for the two molecules in the asymmetric unit for the RGS7 complex and blue for the RGS9 complex) contacts different residues in the RGS domains of RGS7 (orange) and RGS9 (pink). (h) Additional contacts of the RGS7 GGL domain with the RGS domain contribute to a more extensive interface compared to RGS9. The residue differences in RGS9 which cannot form hydrogen bonds are shown in red.

The online version of this article includes the following figure supplement(s) for figure 4:

**Figure supplement 1.** GGL domain interaction with Gβ5 and sequence alignments.

**Figure supplement 2.** Organization of the RGS domain and its interaction with Gβ5.

accommodates Gβ5 with most extensive interactions. As noted before (*Snow et al., 1998*), it is structurally similar to the canonical G protein Gγ1 and Gγ2 subunits as well as GGL domain of RGS9 superimposing with r.m.s.d. values of 1.6 Å, 1.7 Å, and 1.8 Å, respectively (*Figure 4b*). Strikingly, the N-terminal Gα1 α-helix of the RGS7 GGL domain adopts a substantially different conformation. It is tilted about 9.6 Å relative to the position occupied by the corresponding α-helices in all previously reported crystal structures of the Gγ1, Gγ2, and GGL domains of RGS9 (*Figure 4b*). This increases its distance to Gβ5 thus loosing all interactions with the β1 sheet of the WD5 propeller in Gβ5, prominently maintained in RGS9 (*Figure 4c*).

The core of contacts with Gβ5 are formed by residues of the Gα2-Gα5 α-helices of RGS7 GGL (*Figure 4d*, *Figure 4—figure supplement 1a* and *Figure 1—figure supplement 1*). The Gα1-Gα2 loop and the Gα2 α-helix form the most extensive interactions compared to the other α-helices of GGL. Comparison of the GGL-Gβ5 interface between RGS7 and RGS9 shows that they share the majority of Gβ5 contact residues. Yet, two major differences are noted. First, the Gα1 α-helix lost all contacts with Gβ5. Second, replacement contacts are established involving the Gα1-Gα2 loop and the Gα2 α-helix of RGS7 GGL forming hydrogen bonds with Gβ5 to bury 4,918 Å (*Shoichet and Kobilka, 2012*) of solvent accessible surface area (*Figure 4d*, *Figure 4—figure supplement 1a*). These additional contacts are unique to RGS7 and the region involved shows poor sequence conservation with RGS9 GGL (35%), Gγ1 (23%), and Gγ2 (33%) (*Figure 4—figure supplement 1b*). We further detect the 'selectivity filter' within Gα4-Gα5 α-helices that makes the GGL domain incompatible with canonical Gβ binding, similarly to what was seen in RGS9. Overall, these features point to a unique conformational arrangement of Gβ5 positioning it at the heart of interactions within the RGS complex.

## Organization of the RGS domain and its interface with the Gβ5

The C-terminus of the RGS7 molecule contains the hallmark feature of the entire RGS family, namely the RGS homology domain. It is structurally similar to RGS domains of other proteins with known structures composed of nine α-helices (α1-α9) arranged into two bundles. Furthermore, it is nearly identical to the solution structure of the isolated RGS domain of human RGS7 (PDB entry 2d9j) with r.m.s.d. of 0.93 Å, suggesting that the conformation of the RGS domain does not change upon embedding into the context of the entire RGS7-Gβ5 complex (*Figure 4—figure supplement 2a*).

An interesting feature of the RGS domain is its integration with the rest of the complex which occurs through its interactions with the 'bottom' surface of the Gβ5 and GGL domain. The RGS-Gβ5 interface involves two loops (Rα4α5 and Rα6α7) and 3 α-helices (α4, α5, and α7) of RGS7 and 4 loops (S4β1β2, S4β3β4, S5β1β2, and S5β3β4) of Gβ5. They form an extensive network of hydrogen bonds along with hydrophobic/van der Waals interactions that bury 1316 Å of solvent accessible surface area (*Figure 4e*, *Figure 1—figure supplement 1*, and *Figure 4—figure supplement 2b*). Comparison of the RGS domain-Gβ5 interface in RGS7 and the RGS9 dimer structures shows many similarities as well as prominent differences. Most interestingly, electrostatic interactions formed by R375 of α5 and D421 of α7 in RGS7 with E183 and R227 in Gβ5 contribute to a more extensive interface stronger compared to RGS9 (*Figure 4f* and *Figure 4—figure supplement 2b*). The unique rearrangement of these electrostatic interactions in RGS7 likely results in locking the interface at that side thus limiting the flexibility of the RGS domain. Interestingly, the Rα4α5 loop that forms many

direct contacts with Gβ5 is not conserved in the R7 RGS family. A particularly interesting contact point in the RGS domain-Gβ5 interface is formed by the S5β1β2 loop of Gβ5. This loop is longer in Gβ5 compared to other Gβ subunits and shows considerable remodeling of interactions with residues in the RGS domain (*Figure 4g*). It further adopts different conformations between two Gβ5 molecules in the asymmetric unit, resulting in variable interactions with residues of the RGS domain. In particular, T208 in Gβ5 toggles its contact with the RGS7 RGS domain on and off. The corresponding region in RGS9 engages a neighboring residue, E207, which is absent in RGS7. This organization suggests a possibility of this region serving as a flexible conformational switch for the remodeling of RGS domain by interactions with Gβ5.

Finally, the RGS domain of RGS7 also forms interactions with the GGL domain in a manner distinct from that seen in the RGS9 complex structure. Notably, the GGL-RGS interface in RGS7 is built by a network of hydrogen bonds, a feature that is absent in the corresponding interface of RGS9 (*Figure 4h*). Additionally, the connecting loop between the GGL and RGS domains is longer in RGS9 due to two additional amino acids, providing more flexibility for the rearrangement of the RGS domain (*Figure 4—figure supplement 2c*). Collectively, interactions of the RGS domain with the Gβ5 and GGL domains might lead to more intricate integration of RGS domain in the RGS7 complex as compared to RGS9.

## Conformational dynamics of the RGS7-Gβ5 complex

To obtain insights into the conformational dynamics of the RGS7-Gβ5 complex, we first assessed the distribution of crystallographic temperature factors that reflect movement. We found that the temperature values to be at their lowest throughout the Gβ5 structure indicating its conformational rigidity. The distribution varied across structural elements of RGS7 with modest scores in the RGS domains and higher thermal motions in the DEP-DHEX module and the GGL-linker interfacing with the RGS domain suggesting greater conformational flexibility of these regions (*Figure 5a*).

To experimentally validate our findings, we performed HDX-MS measuring the exchange of amide hydrogens with solvent deuterium in the RGS7-Gβ5 complex in solution. The resultant data were plotted onto the crystal structure of the RGS7-Gβ5 dimer complex as a heat map that reflects its conformational dynamics at the baseline (*Figure 5b* and *Figure 5—figure supplement 1* for sequence coverage). We found that the HDX-MS measurements agreed well with the results of our temperature factor analysis while providing additional detail on the conformational flexibility within the RGS7-Gβ5 complex. Rapid exchange with solvent was observed in the DEP-DHEX module indicating its high flexibility. Two loops, Dα1Dα2 (β-hairpin) and Eα1Eα2, showed the most rapid deuterium exchange consistent with the dynamic nature of these regions also captured by the alternate conformations of the two molecules in the asymmetric unit in the crystal (*Figure 2—figure supplement 1b*). The second prominent region with high flexibility included the DHEX-GGL linker as well as α1 and α2 α-helices of GGL that interface with the RGS domain. Flexibility was also noted for the N-terminus of Gβ5. In contrast, both the RGS domain and the core β-propeller fold of Gβ5 exhibited low exchange rates supporting the notion of their conformational rigidity.

Finally, to better understand intra-molecular transitions in RGS7-Gβ5 complex we conducted molecular dynamics (MD) simulations and performed dynamical network analysis (*Eargle and Luthey-Schulten, 2012*; *Sethi et al., 2009*). This analysis partitioned the RGS7-Gβ5 complex into various communities (*Figure 5c*, *Figure 5—source datas 1* and *2*), which represent structural 'mini-domains' or 'substructures' that are highly intraconnected, loosely interconnected, and defined by shared dynamical features. The analysis also identified critical nodes, which are key interactions between the different communities that facilitate dynamical communication between the communities. It revealed that the DHEX and DEP domains each form a distinct community linked by several critical nodes that facilitate communication between the domains. The DEP domain community also includes several loop regions from the Gβ5 β-propeller fold. Gβ5, which is engulfed between the DHEX/DEP domains and the GGL-linker/RGS domain, was partitioned into four communities that split the β-propeller fold WD40 repeat into four communities of nearly equivalent size. The DEP domain and Gβ5 loop community forms a hotspot of critical node interactions with the β-propeller core, indicating this region is important for allosteric dynamical communication between DEP domain and Gβ5. The GGL-linker is part of one of the four Gβ5 β-propeller communities that form critical nodes with the DEP domain and importantly a critical communication node was identified that links the GGL-linker to RGS domain. Thus, the dynamical network analysis revealed a potential

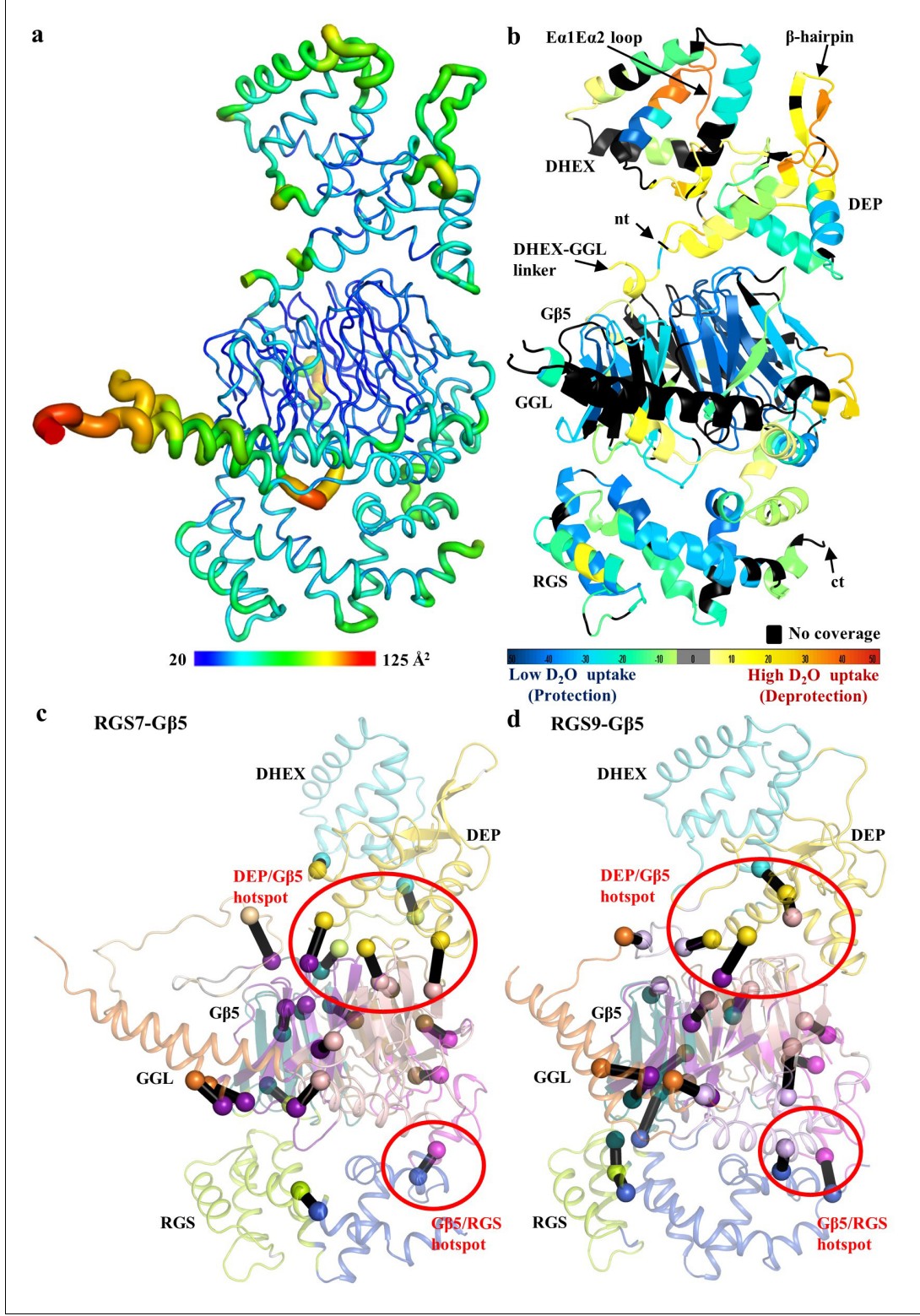

**Figure 5.** Conformational dynamics of the RGS7-Gβ5 dimer. (a) Temperature factor (B-factor) variations plotted on the RGS7-Gβ5 structure as putty cartoon view. Colored as blue to red for low to higher B-factors, respectively. (b) Mapping baseline accessibility of peptides revealed by HDX-MS onto the crystal structure of RGS7-Gβ5. The color coding reflects that the HDX rates are higher (red) reflecting greater flexibility to lower (blue) reflecting slower conformational dynamics. (c, d) Dynamical network analysis of MD simulations performed on (c) RGS7-Gβ5 and (d)

*Figure 5 continued on next page*

*Figure 5 continued*

RGS9-Gβ5 reveals conserved communities or structural 'mini-domains' that have shared dynamical features (color coded using the same colors). And critical interaction nodes calculated between the various communities, which form dynamical signaling interfaces between communities (shown as spheres for Cα atoms, which are color coded by community linked by black lines). Red circles highlight critical nodes that form key hotspots for allosteric communication between the RGS DEP domain and Gβ5; and Gβ5 and the RGS GGL-linker and RGS domain. The online version of this article includes the following source data and figure supplement(s) for figure 5:

**Source data 1.** RGS7-Gβ5 communities.
**Source data 2.** RGS7-Gβ5 critical nodes.
**Source data 3.** RGS9-Gβ5 communities.
**Source data 4.** RGS9-Gβ5 critical nodes.
**Figure supplement 1.** Baseline HDX mapping of RGS7 and Gβ5.

allosteric pathway that links communication between the DHEX/DEP domains and the GGL-linker/RGS domains through Gβ5.

To determine if the putative allosteric pathway linking the DHEX/DEP domains and the GGL-linker/RGS domains through Gβ5 may be shared by other RGS family members, we performed MD simulations and the network community analysis using the previously reported crystal structure of the RGS9-Gβ5 complex (*Cheever et al., 2008*). The computed network communities and critical nodes between communities for RGS9-Gβ5 (*Figure 5d*, *Figure 5—source datas 3* and *4*) are similar to that observed for RGS7-Gβ5. There are some small differences between the two sets of communities; for example the RGS7 DEP community (*Figure 5c*; displayed in yellow) slightly extends into the loop regions of Gβ5, but in the RGS9 complex these loops are part of a core Gβ5 community that slightly extends into the DEP domain (*Figure 5d*; displayed in pink). As such, residues comprising the critical nodes between the DEP domain and the loops connect the Gβ5 β-propeller domain do not involve residues in the exact same position based on a structural alignment. However, the same surfaces defining the critical node connections observed in the RGS7-Gβ5 analysis (*Figure 5c*) are conserved in the RGS9-Gβ5 analysis (*Figure 5d*), including the DEP/Gβ5 hotspot and Gβ5/RGS domain hotspot, suggesting that the potential allosteric pathway may be conserved among other members of RGS family, at least within this analysis of the only two available crystal structures of RGS-Gβ5 complexes.

## Structural insights into modulation of the RGS7-Gβ5 complex by R7BP

One of the key modulators of the RGS7-Gβ5 complex function with critical implications for its physiological regulation is the small SNARE-like molecule R7BP. Therefore, we sought to determine how organization and dynamics of the RGS7-Gβ5 complex is influenced by its association with R7BP. For this purpose, we identified the structural determinants in RGS7-Gβ5 influenced by R7BP binding by differential HDX-MS. In these experiments, we compared the HDX rates and mapped differences in deuterium exchange between the RGS7-Gβ5 dimer and the RGS7-Gβ5-R7BP trimer (*Figure 6a*). Consolidated differential HDX-MS data were mapped onto our structure of the RGS7-Gβ5 dimer and color coded according to the extent of protection that reflects decreased solvent accessibility typically observed upon ligand binding or deprotection associated with increases in protein flexibility (*Figure 6b* and *Figure 6—figure supplement 1* for differential coverage mapping). This differential HDX-MS data revealed robust and highly focused patterns of stabilization induced by the R7BP association. Most prominently, protected regions included an entire DEP domain and adjacent parts of the DHEX (*Figure 6b*), consistent with their involvement in R7BP binding (*Martemyanov et al., 2005*; *Anderson et al., 2007*). Furthermore, more subtle changes were observed in the N-terminus of the Gβ5, the DHEX-GGL linker, and two α-helices of the GGL domain involved in interactions with the RGS domain. Interestingly, the S7β1 sheet and a part of the S6β4-S7β1 loop of Gβ5 that together form a 'conduit' linking the DEP-DHEX and the GGL domains on opposite sides of Gβ5 were also significantly protected from deuterium exchange by R7BP binding, likely by indirect stabilization via a network of interacting regions. No significant changes in deuterium exchange were seen in the RGS domain and throughout most of the Gβ5 core. Importantly, these regions of Gβ5 and RGS7 showing protection from HDX upon binding R7BP comprise critical node regions identified in the molecular dynamics simulations via dynamical network analysis (*Figure 5c*), features which are

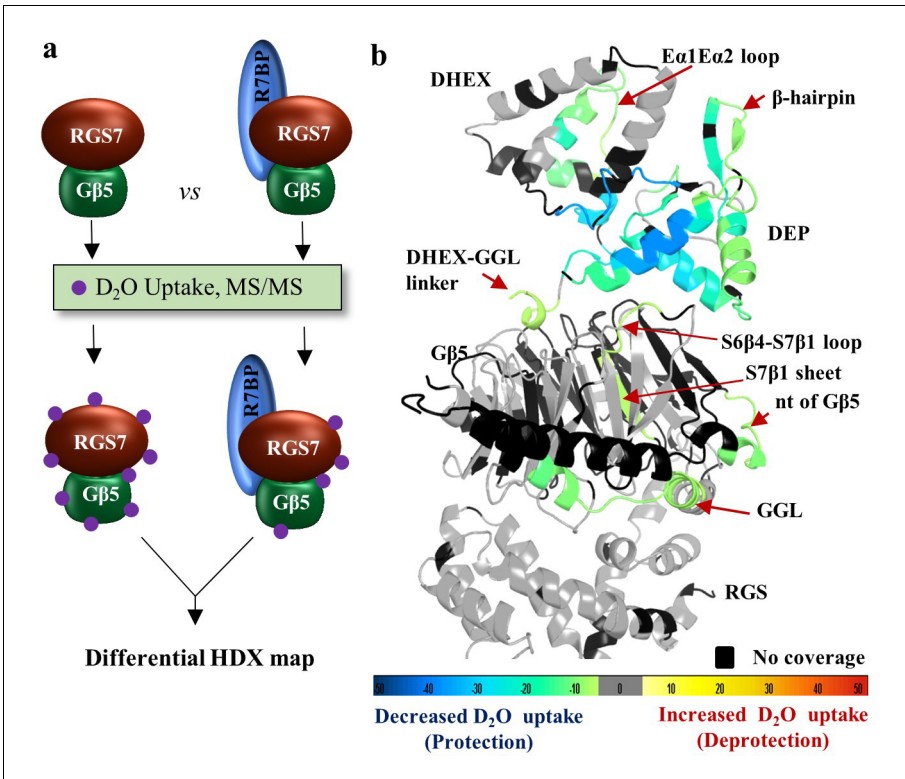

**Figure 6.** R7BP binding interface revealed by HDX-MS. (a) Schematics of the HDX-MS experiment. Three protein samples were explored separately for the HDX-MS analyses and differential HDX data were obtained. Protection from deuterium exchange suggests the stabilization of a region upon protein-protein interaction. (b) Differential consolidation of HDX data are mapped onto the RGS7-Gβ5 dimer structure and shows stabilization of the DEP domain upon R7BP binding by protection from deuterium exchange. Altered conformational dynamics by R7BP binding are indicated by a red arrow.

The online version of this article includes the following figure supplement(s) for figure 6:

**Figure supplement 1.** Differential mapping of changes in HDX of RGS7 and Gβ5 induced by R7BP binding.

shared with the RGS9-Gβ5 complex (*Figure 5d*), validating the allosteric communication pathway between the DEP domain and GGL-linker/RGS domain through the Gβ5 β-propeller core. Together, these findings suggest that binding to R7BP also induces long-distance conformational rearrangements of the RGS7 complex.

## Discussion

The structure of the RGS7 complex and analysis of its conformational dynamics presented here provide critical insights into its mechanisms in regulating GPCR signaling and explains a wealth of biochemical observations. Overall, the structure supports a model of modular organization of the R7 RGS proteins highlighting distinct interfaces that integrate separate elements into a coordinated complex. Our RGS7 structure displays a distinct organization compared to the RGS9 structure.

One of the most striking features of the R7 RGS proteins is their constitutive recruitment of the outlier G-protein β subunit, Gβ5, a central organizational hallmark of the complexes. The present structure shows a unique mode of Gβ5 integration into the complex providing the first example of Gβγ like complexes in which N-terminal α-helices of both Gβ subunit and GGL are protruding away from β-propeller fold of Gβ subunit. These features point to the importance of the RGS-Gβ5 interactions in controlling conformational transitions in the complex. One of the key interfaces formed by Gβ5 involves interaction of its top 'hot-spot' surface with the DEP-DHEX module, which forms a major regulatory hub of the molecule.

The DEP domain is found in many signaling proteins and has been shown to be involved in membrane anchoring and protein-protein interactions with regulatory proteins (*Consonni et al., 2014*; *Paclíková et al., 2017*; *Ballon et al., 2006*) while the DHEX domain is a unique feature of the R7 family. In the R7 RGS proteins, both DEP and DHEX domains are indispensable for binding to membrane anchors R7BP and R9AP (*Anderson et al., 2007*). Our HDX data reveal that the DEP domain indeed forms a major binding site for R7BP that further engages the Gβ5 and the GGL domains. These observations are in good agreement with site-directed mutagenesis data that indicate requirement of both DEP-DHEX and Gβ5 for R7BP/R9AP binding (*Masuho et al., 2011*). Interestingly, organization of the DEP-DHEX module in the RGS7 complex shows marked differences from that seen in RGS9-Gβ5 including a distinct conformation of the β-hairpin and Eα1Eα2 loops, the DHEX orientation relative to DEP and clustering of basic residues on the surface. These differences likely underlie selectivity and strength of the RGS interaction with membrane anchors restricting RGS7 to interact only with R7BP while increasing the affinity of RGS9 for R7BP and allowing it to additionally bind R9AP. One particularly interesting structural feature in this regard is the β-hairpin known for its involvement in protein-protein interactions sensitive to mutations that affect its conformation (*Figure 2—figure supplement 2*; *Wong et al., 2000*; *Yu et al., 2010*). Reminiscent of this, we find that the dynamic β-hairpin of RGS7 is stabilized upon R7BP binding, suggesting it role in binding and likely undergoing conformational rearrangements. Curiously, a pathogenic mutation (R44C) in the β-hairpin of RGS7 that causes melanoma reduces the stability of RGS7 and its catalytic activity, further supporting the importance of conformational transitions in this region in regulating RGS7 function (*Qutob et al., 2018*).

Our MD simulations, HDX data, and comparison of the RGS7 and RGS9 complexes structures suggest that the interaction of the DEP domain with Gβ5 forms a critical hotspot for communication in the complex, which propagates the conformational signal to the GGL-linker and RGS domain. The DEP/Gβ5 interface undergoes large changes in HDX upon R7BP binding that result in shielding constituent residues from solvent accessibility consistent with the stabilization of this region. These observations are consistent with the dynamic nature of the DEP-DHEX-Gβ5 interface organization which was proposed to exist in 'open' and 'closed' states (*Narayanan et al., 2007*). There are several biological implications of this regulation that also explain noted differences between RGS9 and RGS7 complexes.

First, we believe it serves as a major determinant of proteolytic stability of the complexes, which is a major point for controlling their expression levels and resultant negative pressure on GPCR signaling. RGS9 is notably more susceptible to proteolytic degradation than RGS7 and relies to a greater extent on R7BP binding for stability (*Anderson et al., 2009a*). Accordingly, we find that the DEP-Gβ5 interface of RGS7 is more extensive, and thus less dynamic, than that of RGS9. The mutations that weaken the DEP-Gβ5 interface increase susceptibility of the complex to proteolytic degradation (*Porter et al., 2010*). Taken together, this suggests a model whereby R7BP acts to promote the 'closed' conformation, stabilizing the DEP-DHEX interactions with Gβ5 to hide degradation determinants and increase proteolytic stability of the complex.

Second, our model supports a role of the DEP-DHEX-Gβ5 interface in regulating the association of RGS complexes with components of G-protein signaling cascades. The 'hot spot' of the G-protein β subunits is well known to engage in interactions with effectors, modulatory proteins, and Gα subunits. Interestingly, residues involved in effector contacts show high levels of conservation (80–90%) in Gβ5; thus, it is possible that it serves a similar function in docking signaling molecules. While the exact structural determinants mediating this binding are not known, the occlusion of the 'hotspot' interface by the DEP-DHEX domain suggests that the recruitment of R7 RGS complexes to effectors via Gβ5 could be impossible without re-arrangement of the DEP-DHEX module opening up this interface. Thus, modulation of this interface by R7BP/R9AP may play a role in controlling effector association. One notable exception is the Gβ5-mediated association of the RGS7 complex with the GIRK channel, that is promoted by R7BP (*Ostrovskaya et al., 2014*; *Zhou et al., 2012*). Interestingly, based on the crystal structure of the Gβ1-GIRK complex, the analogous surface in Gβ5 possibly involved in GIRK interaction is not occluded by the DEP-DHEX module and RGS7/Gβ5 binding to GIRK could be accommodated with minimal repositioning of cytoplasmic domain of GIRK channel (*Figure 7a*). Comparison of interfaces of RGS7 DEP-Gβ5 and GIRK2-Gβ1γ2 further shows poor conservation of amino acids with only one shared residue (W107 in Gβ5, W99 in Gβ1) (*Figure 3—figure supplement 3* and *Figure 3—figure supplement 1*). This suggests the distinct binding mode of

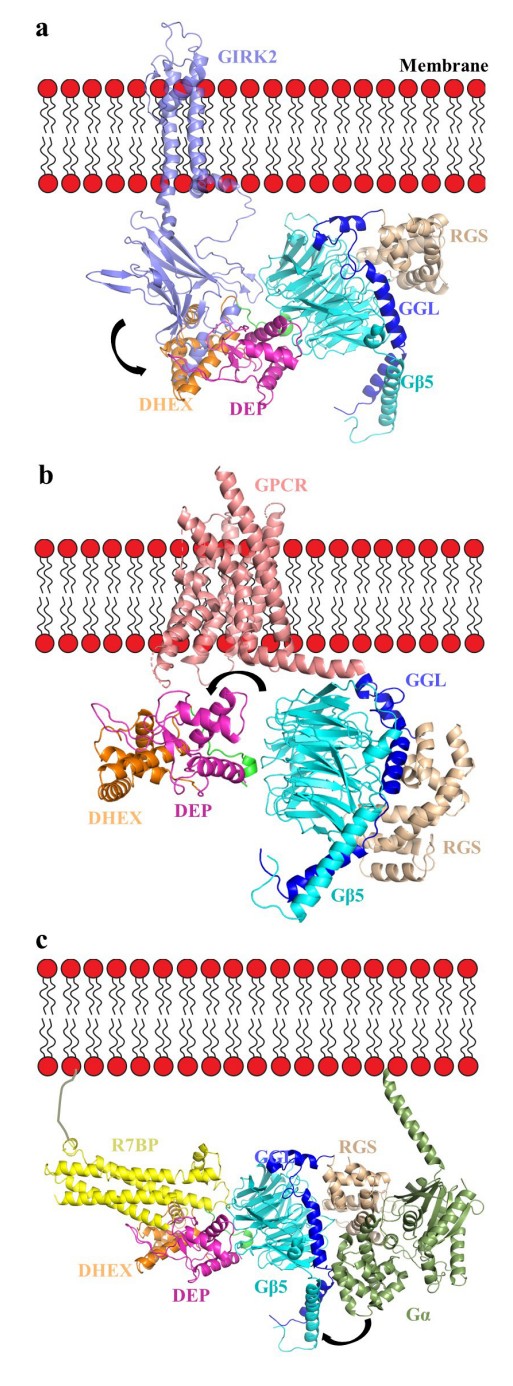

**Figure 7.** Orientation of the RGS7-Gβ5 dimer complex relative to the membrane in complex with GIRK channel, GPCR, and R7BP. (a) A cartoon representation showing the assembly of the GIRK-RGS7-Gβ5 complex. The RGS7-Gβ5 dimer is docked onto the GIRK2-Gβγ complex (PDB entry 4kfm). Gβ5 is docked onto the Gβ1 binding interface of the GIRK channel. The respective GIRK binding site on Gβ5 is not capped by the DEP-DHEX domain however, the other part of cytoplasmic channel domain (CTCD) of GIRK clashes with the DEP-DHEX domain (arrow). (b) A model representing the
*Figure 7 continued on next page*

RGS7 and GIRK2 on Gβ subunits and likely explains why binding of RGS7-Gβ5 does not activate GIRK the way Gβγ association does. The hotspot surface of Gβ5 further features conservation of many residues which in canonical Gβ subunits mediate binding to Gα (*Figure 3—figure supplement 1*). Indeed, Gβ5 has been shown to be interacting with Gα subunits when complexed with Gγ instead of RGS (*Yoshikawa et al., 2000*). Furthermore, RGS9-Gβ5-R9AP complex co-purifies with tightly bound Gα-GDP from endogenous tissues (*Hu and Wensel, 2002*). However, the extensive shielding of the putative Gα binding interface by the DEP-DHEX module makes such association unlikely without 'uncapping' requiring conformational re-arrangement that would swing the DEP-DHEX module away from Gβ5 (*Cheever et al., 2008*). Given our finding that R7BP stabilizes this interface in the RGS7-Gβ5 complex, we suggest that such uncapping would further require the severing of the association with R7BP. Based on these considerations, our model suggests that at the ground state, the hotspot binding surface of Gβ5 in the context of a stable complex with R7BP and RGS7 is inaccessible to Gα-GDP. Thus, association of R7 RGS proteins with Gα-GDP at the hotspot interface, if it occurs, likely requires extensive rearrangement of the complex possibly triggered by yet undiscovered events.

Another intriguing consideration involves the role of the DEP-DHEX-Gβ5 interface in the association with GPCRs. One of the key roles of conventional Gβ subunits is its interfacing with GPCRs and recent crystal structures reveal the mode of the Gβ association with receptors (*Liang et al., 2017*; *Liang et al., 2018*; *Zhang et al., 2017*). The RGS7-Gβ5 complex is also well documented to associate with several GPCRs including the M3 muscarinic receptor and the orphan receptors GPR158 and GPR179. This association is impacted by R7BP (*Orlandi et al., 2012*; *Sandiford and Slepak, 2009*) suggesting involvement of the Gβ5 and DEP-DHEX interface. Interestingly, the amino acids of Gβ subunit involved in GPCR contacts are also reasonably well preserved in Gβ5 (3 out of 5 are conserved) and RGS7-Gβ5 can be confidently docked to accommodate GPCR interactions (*Figure 7b*). However, as in the case of Gα-GDP binding to the Gβ5, docking of RGS7-Gβ5 onto GPCR based on interactions of the canonical Gβγ complexes requires rearranging of the DEP-DHEX module away from the hotspot surface of Gβ5. This fuels a tempting speculation that interactions with GPCR may in fact provide the conformational

*Figure 7 continued*

GPCR-RGS7-Gβ5 assembly. The RGS7-Gβ5 dimer is docked onto GPCR similar to Gβγ in a typical GPCR-Gαβγ complex (here we used calcitonin receptor-heterotrimeric Gαs protein complex structure; PDB entry 5uz7). The DEP-DHEX domain (shown as arrow) is repositioned as it capped the probable GPCR binding interface of the Gβ5 subunit. (c) A model representing the tetrameric assembly complex of RGS7-Gβ5-R7BP-GαGTP. The complex is oriented relative to membrane according to the Gα position and its interaction with the RGS domain as well as docked to R7BP near its binding site based on HDX analysis. Structural rearrangement is required to avoid clashes of the helical domain of Gα with RGS7 GGL and Gβ5 (shown in arrow).

The online version of this article includes the following figure supplement(s) for figure 7:

**Figure supplement 1.** Interaction of the RGS domain with Gβ5 and probable interface of RGS-Gα.
**Figure supplement 2.** Homology model of R7BP.

trigger for the re-arrangement in RGS7-Gβ5-R7BP complex that leads to the opening of the interface between DEP-DHEX and the hotspot surface of Gβ5 much in the same way GPCRs promote swinging of α-helical domain in Gα (*Rasmussen et al., 2011*).

Third, a major implication of dynamic interfacing of Gβ5 with multiple domains of RGS7 concerns regulation of the catalytic activity on its Gα-GTP substrates. To better understand the possible mechanisms, we modeled the association of RGS7-Gβ5 with activated Gα by superposing the structures of RGS7-Gβ5 and the RGS9 RGS domain-Gαt/i1 complex (PDB entry 1fqk; *Figure 7—figure supplement 1a*). The Gα subunit docks well and supports the canonical interaction of Rα3-α4, Rα5-α6, and Rα7-α8 of RGS7 directly with switch I and II residues of Gα observed in several many RGS-Gα complexes (*Figure 7—figure supplement 1b, c*). These loops of the RGS domain move closer to switch regions to facilitate the stabilization of the Gα transition state and increases GTP

hydrolysis (*Slep et al., 2001*). Comparison of Gα binding residues of RGS9 with RGS7 shows conservation of residues with minor variations (*Figure 7—figure supplement 1d*). However, the αB-αC residues of the helical domain sterically clash with residues of the GGL α−helices and the N-terminus of Gβ5 (*Figure 7—figure supplement 1a*). These clashes are consistent with noted inhibitory effects of Gβ5 on binding of the RGS domain to activated Gα (*Levay et al., 1999*) and suggest that recognition of Gα-GTP by RGS7-Gβ5 could be improved by structural rearrangement of the RGS domain relative to Gβ5. The HDX-MS analysis shows stabilization of the region (GGL α-helices and the N-terminus of Gβ5) clashing with Gα residues (*Figure 6b* and *Figure 7—figure supplement 1a*) suggesting that conformational rearrangement of this region upon R7BP binding could alleviate this steric hindrance and improve Gα binding. When considered with our MD simulations, these data suggest that R7BP binding exerts its effects via remodeling contacts initiated at the top surface of Gβ5 which are translated by the S7β1 blade of the WD6 β-propeller to the opposite side and impact the network of the RGS domain contacts with the Gβ5 to allow changes in their relative orientations. Thus, Gβ5 may serve as a conduit of the conformational changes triggered at the N-terminus of the molecule to impact the organization of the RGS domain at the C-terminus of the RGS7 molecule. This model could explain biochemical observations that the GAP activity of RGS7-Gβ5 can be markedly enhanced upon association with R7BP (*Masuho et al., 2013*; *Drenan et al., 2006*). Although membrane recruitment of RGS7-Gβ5 by itself plays a major role in augmenting the catalytic activity of RGS7-Gβ5 (*Muntean and Martemyanov, 2016*), rearrangement of the RGS domain triggered by R7BP likely further contributes to fine tuning this process, possibly in coordination with GPCRs.

To obtain further insights, we modeled R7BP based on its homology with SNARE proteins (PDB entries 1fio, 2xhe, 1hs7). An iterative structural assembly simulation by the I-TASSER algorithm predicts R7BP to form a four-helix bundle (*Figure 7—figure supplement 2*). The resulting model was docked onto the RGS7-Gβ5 structure guided by our HDX data (*Figure 7c*). We further docked the Gα−GTP onto the RGS domain guided by the structure of the RGS domain of RGS9 in complex with the Gαt/i1 chimera (PDB entry 1fqk). The resulting model illustrates the possible orientation of the tetramer on the membrane constrained by the positively charged patch in DEP-DHEX, C-terminal palmitoylation on R7BP, and N-terminal lipidation of Gα (*Figure 7c*). Notably, the orientation of the complex involving R7BP on the membrane is also compatible with the predicted macromolecular assemblies involving the GPCR and the GIRK.

In summary, our study reveals the versatile nature underlying the organization of the macromolecular complex that exploits several interfaces and mechanisms for interaction with GPCR signaling

components undergoing conformational transitions that adjust its function allowing it to serve its critical needs in precise regulation of key neuro-modulatory reactions in the nervous system.

# Materials and methods

**Key resources table**

| Reagent type (species) or resource | Designation | Source or reference | Identifiers | Additional information |
|---|---|---|---|---|
| Genetic reagent (baculovirus) | His-RGS7 | This Paper | N/A | |
| Genetic reagent (baculovirus) | Gβ5 | This Paper | N/A | |
| Genetic reagent (baculovirus) | GST-PP-R7BP | This Paper | N/A | |
| Cell line (insect cells) | Sf9 | ThermoFisher Scientific | 11496015 | |
| Chemical compound, drug | Sypro Orange dye | Sigma-Aldrich | S5692 | |
| Chemical compound, drug | Crystallization plates | TTP Labtech | 4150–0560 | |
| Chemical compound, drug | Crystallization screens | Hampton Research and Qiagen | N/A | |
| Chemical compound, drug | Polyethylene glycol 3350 | Sigma | 88276–250G | |
| Chemical compound, drug | TCEP-HCl | Hampton Research | HR2-801 | |
| Software, algorithm | MASCOT | Matrix Science | N/A | |
| Software, algorithm | HDX Workbench | *Pascal et al., 2012* | PMID: 22692830 | |
| Software, algorithm | autoPROC | *Vonrhein et al., 2011* | https://www.globalphasing.com/autoproc/ | |
| Software, algorithm | PHASER | *McCoy et al., 2007* | http://www.phaser.cimr.cam.ac.uk/index.php/ | |
| Software, algorithm | PHENIX | *Adams et al., 2010* | https://www.phenix-online.org | |
| Software, algorithm | BUSTER | *Bricogne, 2011* | https://www.globalphasing.com/buster/ | |
| Software, algorithm | Coot | *Emsley and Cowtan, 2004* | http://www2.mrc-lmb.cam.ac.uk/personal/pemsley/coot/ | |
| Software, algorithm | MolProbity | *Chen et al., 2010* | http://molprobity.biochem.duke.edu | |
| Software, algorithm | PyMOL | Schršdinger, LLC | https://pymol.org/2/ | |
| Software, algorithm | I-TASSER | *Yang and Zhang, 2015* | https://zhanglab.ccmb.med.umich.edu/I-TASSER/ | |
| Software, algorithm | PISA | N/A | http://www.ebi.ac.uk/msd-srv/prot_int/cgi-bin/piserver | |
| Software, algorithm | Modeller | *Eswar et al., 2006* | PMID: 18428767 | |

*Continued on next page*

*Continued*

| Reagent type (species) or resource | Designation | Source or reference | Identifiers | Additional information |
|---|---|---|---|---|
| Software, algorithm | AMBER 16 | AMBER software, University of California, San Francisco | http://ambermd.org/ | |
| Software, algorithm | H ++ server | *Gordon et al., 2005* | http://biophysics.cs.vt.edu/H++ | |
| Software, algorithm | UCSF Chimera | *Pettersen et al., 2004* | http://www.rbvi.ucsf.edu/chimera/ | |
| Software, algorithm | cpptraj | *Roe and Cheatham, 2013* | PMID: 26583988 | |
| Software, algorithm | VMD | *Humphrey et al., 1996* | https://www.ks.uiuc.edu/Research/vmd/ | |
| Software, algorithm | Carma | *Glykos, 2006* | https://utopia.duth.gr/glykos/Carma.html | |
| Software, algorithm | CatDCD | Written by Justin Gullingsrud | http://www.ks.uiuc.edu/Development/MDTools/catdcd | |
| Software, algorithm | gncommunities | By Luthey-Schulten group | http://faculty.scs.illinois.edu/schulten/software/networkTools/index.html | |

## Expression and purification of the RGS7-Gβ5 complex

The RGS7-Gβ5 complex was co-expressed and purified as described previously (*Martemyanov et al., 2005*; *Muntean et al., 2018*) with modifications. Briefly, RGS7 was co-expressed with Gβ5 in Sf9 insect cells via the baculovirus-mediated expression system. Insect cells containing the recombinant RGS7-Gβ5 complex were lysed in buffer A (20 mM HEPES pH 8, 300 mM NaCl, 10 mM imidazole (pH 8), 5 mM β-mercaptoethanol, 1% (v/v) glycerol) containing EDTA-free complete protease inhibitor tablets (Roche) by sonication. The recombinant RGS7-Gβ5 complex containing supernatant was clarified by centrifugation at 32,000 rpm for 30 min and loaded onto pre-equilibrated HisTALON Superflow Cartridge (Clontech Laboratories, Inc) with buffer A. The protein complex was eluted over a 250 mM imidazole gradient. The protein complex was analyzed by SDS-PAGE and fractions containing the RGS7-Gβ5 complex were pooled, dialyzed, and loaded onto a MonoQ column and eluted over a 1 M NaCl gradient. The eluted complex was further purified by using a Hiload 26/60 Superdex 200 column (GE Healthcare), which was pre-equilibrated with buffer B (20 mM PIPES pH 6.5, 200 mM NaCl, 2 mM TCEP-HCl). Peak fractions were analyzed by SDS-PAGE, pooled, and concentrated to 18 mg/ml.

## Crystallization of the complex

We performed extensive crystallization trials for the RGS7-Gβ5 dimer complex using several commercially available crystallization screens (~1000 conditions) by vapor diffusion hanging drop at 20°C and 4°C. We obtained only poorly diffracting crystals that we further optimized at 4°C and obtained crystals that diffracted X-rays beyond 2.0 Å Bragg spacings from a reservoir solution of 0.1M sodium malonate (pH 6.25) and 10% w/v Polyethylene glycol 3350 over 3–4 days. Crystals were harvested by flash-freezing directly into liquid nitrogen by including 30% ethylene glycol as a cryoprotectant.

## X-ray structure determination, model building, and crystallographic refinement

X-ray diffraction data were collected on beamline 22-ID of SER-CAT at the Advanced Photon Source at Argonne National Laboratory. All X-ray diffraction data were processed and integrated and scaled using XDS and AIMLESS as implemented in autoPROC package (*Vonrhein et al., 2011*). A 2.13 Å dataset was obtained. The space group was determined to be $P2_1$ (space group number 4). Molecular replacement was performed using the program Phaser in the CCP4 package (*McCoy et al., 2007*) with the RGS9-Gβ5 structure (PDB entry 2pbi) as the search model. Two molecules in the

asymmetric unit were identified. Iterative model-building was performed using COOT (*Emsley and Cowtan, 2004*) and refined by performing maximum likelihood as implemented in autoBUSTER (*Bricogne, 2011*) and PHENIX (*Adams et al., 2010*). The quality of the structure was assessed using MolProbity (*Chen et al., 2010*). Data processing and structure refinement statistics for isotropic (to 2.81 Å) and anisotropic (to 2.13 Å) are provided in *Table 1*. Structural figures and electrostatic potential were generated using PyMOL (https://pymol.org/2/). Accessible surface area (ASA) were obtained using 'Protein interfaces, surfaces and assemblies' service PISA at the European Bioinformatics Institute (http://www.ebi.ac.uk/pdbe/prot_int/pistart.html) (*Krissinel and Henrick, 2007*).

## Expression and purification of R7BP

The coding sequence of *Mus musculus* R7BP (residues 47–231) was amplified by PCR. A glutathione S-transferase (GST) tag followed by PreScission Protease site is present at the N-terminus of R7BP. The construct was generated using the baculodirect system (Invitrogen) and expressed into the baculovirus expression system. Insect cells containing recombinant R7BP were harvested, resuspended in buffer C (20 mM Tris pH 8, 300 mM NaCl, 1 mM DTT, 1% (v/v) glycerol) having complete protease inhibitor tablets (Roche), disrupted by sonication and clarified by centrifugation at 32,000 rpm for 30 min. The cell lysate was loaded onto a pre-equilibrated GSTPrep FF 16/10 (GE Healthcare) with buffer C and eluted with 15 mM glutathione. The eluted protein was analyzed by SDS-PAGE and fractions containing R7BP were pooled, dialyzed, and treated by PreScission Protease to cleave off the GST tag. The cleaved tag was removed by further passing the protein over the GST column. The flow-through fraction containing R7BP was concentrated and further purified using a Hiload 26/60 Superdex 75 column (GE Healthcare), which was pre-equilibrated with buffer B. The purity of the protein was analyzed by SDS-PAGE, pooled, and concentrated to 10 mg/ml.

## Formation of the RGS7-Gβ5-R7BP trimer complex

The RGS7-Gβ5 dimer complex and R7BP were mixed together at a 1:1.2 molar ratio and incubated on ice at up to 2 hr. The trimer was loaded onto a Hiload 26/60 Superdex 200 column (GE Healthcare) which was pre-equilibrated with buffer B. The peak fraction containing the trimer complex was analyzed by SDS-PAGE, pooled, and concentrated for crystallization.

## Thermoflour/DSF analysis

A thermofluor shift assay was used to optimize the buffer compositions that could stabilize the complex for structural study. The stability is determined by the $T_m$ (melting temperature) of the protein. The assay was carried out in a 96-well thin-wall PCR plate (BioRad), and the plates were sealed with Optical-Quality Sealing Tape (Bio-Rad). The standardization of the assay was performed using various concentrations of the ratio of the protein complex and the Sypro Orange dye to get ideal peaks to calculate $T_m$. Buffers (pH range 4.0–9.0), NaCl concentration (0–500 mM), reducing agent (DTT and TCEP-HCl), cheating agent (EDTA), stabilizer (glycerol, NDSB-256) and other additives and salts ($MgCl_2$, $CaCl_2$, KCl) were screened. The total reaction volume used was 20 µl and the reactions were heated in a real-time PCR machine (Roche lightcycler 480) from 20°C to 90°C with an acquisition mode of continuous, and 10 acquisitions per degree C. The final protein concentration of protein used was 1.5 µM in a buffer containing 20 M HEPES pH 8.0, 300 mM NaCl, 1 mM DTT, and 3X Sypro Orange dye. We ran the protocol as per LightCycler 480 instrument instructions which is created for the SYPRO Orange format. We obtained reasonably shaped curves with a good amount of fluorescence for calculation of $T_m$ and used the $T_m$ calling analysis feature to obtain negative first derivative drawings of the curves. Optimized buffers for RGS7 complexes were used for crystallization trials and biophysical characterizations. We have observed that pH of buffer has greatest effect on stability of the complex followed by stronger reducing agent TCEP-HCl.

## Size exclusion chromatography with multi-angle light scattering (SEC-MALS)

The purified RGS7 trimer complex in optimized buffer was analyzed by SEC-MALS using an HPLC system (Agilent Technologies 1260 Infinity) equipped with a MALS system (Wyatt DAWN HELEOS II Ambient with Optilab TrEX HC differential refractive index detector). The SEC–MALS system was calibrated with bovine serum album prior to RGS7 complex. The gel filtration purified complex was

loaded onto pre-equilibrated analytical superdex200 (GE Healthcare) with buffer B. An aliquot of 100 µl sample of RGS7 trimer complex at a concentration of 1.2 mg/ml was injected with a flow rate 0.5 ml/min. ASTRA (Wyatt) Windows-based software was used for data collection and analysis.

## Hydrogen-deuterium exchange (HDX) detected by mass spectrometry (MS)

Differential HDX-MS experiments were conducted as previously described with a few modifications (*Chalmers et al., 2006*).

*Peptide Identification:* Peptides were identified using tandem MS (MS/MS) with an Orbitrap mass spectrometer (Q Exactive, ThermoFisher). Product ion spectra were acquired in data-dependent mode with the top five most abundant ions selected for the product ion analysis *per* scan event. The MS/MS data files were submitted to Mascot (Matrix Science) for peptide identification. Peptides included in the HDX analysis peptide set had a MASCOT score greater than 20 and the MS/MS spectra were verified by manual inspection. The MASCOT search was repeated against a decoy (reverse) sequence and ambiguous identifications were ruled out and not included in the HDX peptide set.

## HDX-MS analysis

5 µl of each protein complex (10 µM) was diluted into 20 µl D2O buffer (20 mM Tris-HCl, pH 7.4; 150 mM NaCl; 2 mM DTT) and incubated for various time points (0, 10, 60, 300, and 900 s) at 4°C. The deuterium exchange was then slowed by mixing with 25 µl of cold (4°C) 3 M urea and 1% trifluoroacetic acid. Quenched samples were immediately injected into the HDX platform. Upon injection, samples were passed through an immobilized pepsin column (2 mm × 2 cm) at 200 µl min−one and the digested peptides were captured on a 2 mm × 1cm C8 trap column (Agilent) and desalted. Peptides were separated across a 2.1 mm × 5cm C18 column (Hypersil Gold, ThermoFisher) with a linear gradient of 4–40% CH3CN and 0.3% formic acid, over 5 min. Sample handling, protein digestion and peptide separation were conducted at 4°C. Mass spectrometric data were acquired using an Orbitrap mass spectrometer (Exactive, ThermoFisher). The intensity weighted mean m/z centroid value of each peptide envelope was calculated and subsequently converted into a percentage of deuterium incorporation. This was accomplished by determining the observed averages of the undeuterated and fully deuterated spectra using the conventional formula described elsewhere (*Zhang and Smith, 1993*). Corrections for back-exchange were made on the basis of an estimated 70% deuterium recovery, and accounting for the known 80% deuterium content of the deuterium exchange buffer.

*Data Analysis*: HDX analyses were performed in triplicate, with single preparations of each purified protein/complex. Statistical significance for the differential HDX data is determined by t test for each time point, and is integrated into the HDX Workbench software (*Pascal et al., 2012*).

*Data Rendering*: Deuterium uptake for each peptide is calculated as the average of % D for all on-exchange time points and the difference in average %D values between the apo and ligand bound samples is presented as a heat map with a color code given at the bottom of each figure (warm colors for deprotection and cool colors for protection). Peptides are colored by the software automatically to display significant differences, determined either by a > 5% difference (less or more protection) in average deuterium uptake between the two states, or by using the results of unpaired t-tests at each time point (p-value < 0.05 for any two time points or a p-value < 0.01 for any single time point). Peptides with non-significant changes between the two states are colored grey. The exchange at the first two residues for any given peptide is not colored.

## Homology modeling of R7BP

The three-dimensional structure of mouse R7BP (residues 47–231) was generated using the automated I-TASSER web service (http://zhanglab.ccmb.med.umich.edu/I-TASSER). I-TASSER generates 3D atomic models from multiple threading alignments and iterative structural assembly simulations. It uses multiple PDBs and accuracy of the models was estimated using parameters such as C-score, TM-score, and RMSD (*Roy et al., 2010*; *Yang and Zhang, 2015*). The R7BP homology model is used for docking with RGS7-Gβ5 dimer structure based on HDX-MS data and documented the involvement of both DEP-DHEX and Gβ5 in the trimer complex formation.

## Molecular dynamics simulations

The Modeller (*Eswar et al., 2006*) extension within UCSF Chimera (*Pettersen et al., 2004*) was used to fill in the missing DHEX-GGL linker (R201–P253) of the RGS7-Gβ5 complex structure (chains A and C) to prepare input files for all-atom molecular simulations using AMBER 16. The RGS9-Gβ5 complex structure (PDB: 2PBI) was used without additional modification. The resulting complexes were submitted to H ++ server (*Gordon et al., 2005*) to determine the protonation states of titratable residues at pH 7.4. AMBER names were assigned to different protonation states of histidine using pdb4amber provided in AmberTools 16. Tleap was used to generate topology and coordinate files using the ff14SB force field following a method provided in the AMBER tutorial (http://ambermd.org/tutorials/basic/tutorial5/). The resulting structures were solvated in a truncated octahedral box of TIP3P water molecules with the 10 Å spacing between the protein and the boundary, and neutralized with $K^+$ and $Cl^-$ ions added to 50 mM (the number of iones to add was calculated using a molarity.perl script provided by Prof. Thomas Cheatham III; http://archive.ambermd.org/200907/0332.html). The system was minimized and equilibrated in nine steps at 310 K with non-bonded cutoff of 8 Å. In the first step the heavy protein atoms were restrained by a spring constant of 5 kcal mol$_{-1}$ Å- for 2000 steps, followed by 15 ps simulation under NVT conditions with shake, then two rounds of 2000 cycles of steepest descent minimization with 2 and 0.1 kcal mol$^{-1}$ Å-restraints were performed. After one round without restraints, three rounds of simulations with shake were conducted for five ps, 10 ps and 10 ps under NPT conditions and restraints of 1, 0.5 and 0.5 kcal mol$^{-1}$ Å$^{-2}$ on heavy atoms. Finally, an unrestrained NPT simulation was performed for 200 ps. Production runs were carried out with hydrogen mass repartitioned (*Hopkins et al., 2015*) parameter files to enable 4 fs time steps. Constant pressure replicate production runs were carried out with independent randomized starting velocities. Pressure was controlled with a Monte Carlo barostat and a pressure relaxation time (taup) of 2 ps. Temperature was kept constant at 310 K with Langevin dynamics utilizing a collision frequency (gamma_ln) of 3 ps$^{-1}$. The particle mesh ewald method was used to calculate non-bonded atom interactions with a cutoff (cut) of 8.0 Å. SHAKE (*Ryckaert et al., 1977*) and hydrogen mass repartitioning was used to allow longer time steps. Production simulations were run for 2 μs each for the RGS7-Gβ5 and RGS9-Gβ5 complexes, which were initially analyzed using cpptraj (*Roe and Cheatham, 2013*).

Dynamic network analysis was performed on the last half (1 μs) of the production simulations following methods described in the 'Dynamical Network Analysis' tutorial developed by the Luthey-Schlten Group (http://faculty.scs.illinois.edu/schulten/tutorials/), which is provided as a PDF file (http://faculty.scs.illinois.edu/schulten/tutorials/network/network_tutorial.pdf). Briefly, the analysis required installation of the following software: the molecular visualization program VMD (*Humphrey et al., 1996*) (http://www.ks.uiuc.edu/Research/vmd/), which includes the NetworkView (*Eargle and Luthey-Schulten, 2012*) extension; the molecular dynamics analysis program Carma (http://utopia.duth.gr/glykos/Carma.html) (*Glykos, 2006*); a program that concatenates DCD trajectory files into a single DCD file called CatDCD (http://www.ks.uiuc.edu/Development/MDTools/catdcd/); and the program gncommunities (http://faculty.scs.illinois.edu/schulten/software/networkTools/index.html), which generates network communities from the simulation trajectory using methods described previously by the authors of the program (*Sethi et al., 2009*; *Alexander et al., 2010*; *Black Pyrkosz et al., 2010*). AMBER trajectory and prmtop files from the RGS7-Gβ5 and RGS9-Gβ5 complex simulations were converted to DCD and PSF format, respectively, using the commands 'animate write dcd' and 'writepsf' within VMD, respectively. The dynamic network models were created using the 'networkSetup' command within VMD, which invokes CatDCD and Carma and takes as input PSF and DCD files, information about node selection (Cα atoms were specified), and restrictions that specify constraints on the network generation (notSameResidue and notNeighboringCAlpha were specified); otherwise, default parameters were used for the calculation, that is the contact cutoff distance (distanceCutoff, = 4.5 Å) for at least 75% (requiredOccupancy = 0.75) of the MD trajectory. The resulting output file (contact.dat) was read into VMD using the NetworkView plugin to visualize the computed network. The dynamical network generated by the networkSetup command in VMD was used as input into the program gncommunities, which was invoked within VMD and subsequently read using the NetworkView plugin for initial analysis. The gncommunities program uses a Girvan-Newman algorithm (*Girvan and Newman, 2002*) to partition the dynamical network into subnetworks or 'communities', identifies critical nodes (i.e. residue contacts) that lie in the interface

between two different communities, and determines a 'betweenness' score representing how important the contact is to the entire dynamical network. The resulting output file (communities.out) from the gncommunities calculation contains the list of residues in each community, the critical nodes between communities, and the betweenness score; this information was used to visualize the communities (color coded similarly for the RGS7-Gβ5 and RGS9-Gβ5 complexes) and display the computed critical nodes (we displayed nodes with the largest betweenness scores comprising the top two-thirds of all betweenness values computed, which represent the most critical connections) as spheres with black lines linking two Cα atoms in the PyMOL Molecular Graphics System (Version 2.0, Schrödinger, LLC).

## Acknowledgements

We thank to the staff of the Southeast Regional Collaborative Access Team (SER-CAT) for synchrotron support. We are indebted to Krishna Chinthalapudi (The Scripps Research Institute) for help in general with crystallographic software and Dr. Andrew Ward and Lauren Holden for discussions. This work was supported by NIH Grants DA036596, EY018139, DA042746 (to KAM), GM114420 (to DJK), DK105825 (to PRG) and start-up funds provided to The Scripps Research Institute (to TI)

## Additional information

### Funding

| Funder | Grant reference number | Author |
|---|---|---|
| National Institute on Drug Abuse | DA036596 | Kirill A Martemyanov |
| National Eye Institute | EY018139 | Kirill A Martemyanov |
| National Institute on Drug Abuse | DA042746 | Kirill A Martemyanov |
| National Institute of General Medical Sciences | GM114420 | Douglas J Kojetin |
| National Institute of Diabetes and Digestive and Kidney Diseases | DK105825 | Patrick R Griffin |

The funders had no role in study design, data collection and interpretation, or the decision to submit the work for publication.

### Author contributions

Dipak N Patil, Data curation, Formal analysis, Visualization, Writing—original draft; Erumbi S Rangarajan, Data curation, Formal analysis, Methodology; Scott J Novick, Data curation, Formal analysis, Investigation; Bruce D Pascal, Software, Formal analysis; Douglas J Kojetin, Data curation, Formal analysis, Investigation, Methodology, Writing—review and editing; Patrick R Griffin, Data curation, Formal analysis, Validation, Investigation, Visualization, Writing—review and editing; Tina Izard, Data curation, Formal analysis, Supervision, Project administration, Writing—review and editing; Kirill A Martemyanov, Conceptualization, Funding acquisition, Writing—original draft, Project administration, Writing—review and editing

### Author ORCIDs

Dipak N Patil (ID) http://orcid.org/0000-0003-1112-5557
Douglas J Kojetin (ID) http://orcid.org/0000-0001-8058-6168
Kirill A Martemyanov (ID) http://orcid.org/0000-0002-9925-7599

### Decision letter and Author response

Decision letter https://doi.org/10.7554/eLife.42150.sa1
Author response https://doi.org/10.7554/eLife.42150.sa2

## Additional files

### Supplementary files
• Transparent reporting form

### Data availability

Coordinate and structure factor have been deposited in the protein data bank with accession codes 6N9G. Raw HDX data are deposited at https://doi.org/10.6084/m9.figshare.7316462.v2.

The following datasets were generated:

| Author(s) | Year | Dataset title | Dataset URL | Database and Identifier |
|---|---|---|---|---|
| Patil DN, Rangarajan ES, Izard T, Martemyanov KA | 2018 | Crystal Structure of RGS7-Gbeta5 dimer | http://www.rcsb.org/structure/6N9G | Protein Data Bank, 6N9G |
| Patil DN, Rangarajan ES, Novick SJ, Pascal BD, Kojetin DJ, Griffin PR, Izard T, Martemyanov KA | 2018 | Mass Spec Raw File Data | https://doi.org/10.6084/m9.figshare.7316462.v2 | figshare, 10.6084/m9.figshare.7316462.v2 |

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
