## [Decision Letter]

Thank you for submitting your article "Structural organization of a major neuronal G protein regulator, the RGS7-Gβ5-R7BP complex" for consideration by *eLife*. Your article has been evaluated by two peer-reviewers and by José Faraldo-Gómez as Reviewing Editor and Richard Aldrich as the Senior Editor. Both peer-reviewers have agreed to reveal their identity: Gregory R Bowman (Reviewer #1); Stephen R Sprang (Reviewer #2).

The reviewers have discussed the reviews with one another and with the Reviewing Editor, who has drafted this decision to help you prepare a revised submission.

Summary:

Patil et al., describe the X-ray structure of the RGS7:Gb5 complex, along with molecular dynamics simulations and hydrogen-deuterium exchange data and some analysis of the interaction with R7BP. This work complements an earlier study of the RGS9:Gb5 complex by Sondak et al., by revealing structural differences in the interaction of two R7-family RGS homologs with Gb5, the work contributes to explain the factors determining the specificity of this interaction. A number of interesting observations and mechanistic speculations are presented on the basis of the combined structural and dynamic analysis of RGS7. Altogether, this is a welcome contribution to the literature, in light of the rather limited structural data available for R7-family proteins.

Essential revisions:

1) The presentation of the results from MD analysis needs improving. What criteria were used to delimit or define the residue sets that constitute different "communities"? Simply citing Eargle and Luthey-Schulten, 2012 and Sethi et al., 2009 to "dynamic network analysis" without any discussion of how this procedure was undertaken is not sufficient. What software was used? What were the parameters used? The analysis should be described in sufficient detail to be reproducible by reasonably skilled structural biologist. It is also not clear how "nodes" are defined. Figure 5C is noisy and confusing. Consider color-coding the different "communities".

2) Regions where RGS7 differs most from RGS9 also appear to be more dynamic, based on the HDX data and B-factors. It is therefore a concern whether these structural differences are mechanistically significant, or whether the crystal structures merely capture alternative conformations of flexible regions of the protein. In regard to the dynamic network analysis, it is unclear whether the observations made for RGS7 are a specific feature of this protein and/or the structure simulated. Simulations of RGS9:G5b ought to be carried out to more clearly discern the structural/dynamical similarities/differences between RGS7:Gb5 and RGS9:G5b. Analysis of the dynamic networks within RGS9:G5b should also be carried out to establish whether the results obtained for RGS7 can be extended to other members of the R7 family.

---

## [Author Response]

1) The presentation of the results from MD analysis needs improving. What criteria were used to delimit or define the residue sets that constitute different "communities"? Simply citing Eargle and Luthey-Schulten, 2012 and Sethi et al., 2009 to "dynamic network analysis" without any discussion of how this procedure was undertaken is not sufficient. What software was used? What were the parameters used? The analysis should be described in sufficient detail to be reproducible by reasonably skilled structural biologist. It is also not clear how "nodes" are defined. Figure 5C is noisy and confusing. Consider color-coding the different "communities".

We appreciate these comments in response to which we focused on detailing the procedures we used for the dynamical network analysis in a more transparent way. We updated the methods section to include a new paragraph with detailed information about this analysis, web hyperlinks to tutorials and other information that we used to perform the analysis, and we list all of the software used for the analysis along with any pertinent references. Our revised methods should contain sufficient detail such that other researchers can reproduce the analysis, including information about the specific parameters used in the analyses. We also briefly describe how “communities” and “critical nodes” are calculated and defined; however, we provide citations to literature describing the development of the methodology we used, which includes detailed information about the methodology from the authors of the methodology, since we did not develop the methodology ourselves.

Regarding Figure 5C, we previously showed all computed nodes (as lines connecting Cα atoms) within the RGS7-GB5 complex (which made the figure panel overly complex) and colored the lines according to their respective communities. In our revised manuscript, instead of color-coding the lines connecting all of the computed nodes, we color-coded a cartoon diagram of the complex without showing the lines connecting all of the computed noes. We also used a better color palette, which makes discerning the various communities within each complex easier, and we used the same color-coding palette for each of the shared communities in the RGS7-GB5 complex (new Figure 5C) and RGS9-GB5 complex (new Figure 5D). Because this new method of displaying the communities as a cartoon diagram is considerably less noisy, we were able to display the critical nodes within the same figure panel (shown as spheres for Cα atoms connected by black lines); previously we separated this into two figure panels (C and D) due to the overly complex nature of our old figure panels.

2) Regions where RGS7 differs most from RGS9 also appear to be more dynamic, based on the HDX data and B-factors. It is therefore a concern whether these structural differences are mechanistically significant, or whether the crystal structures merely capture alternative conformations of flexible regions of the protein. In regard to the dynamic network analysis, it is unclear whether the observations made for RGS7 are a specific feature of this protein and/or the structure simulated. Simulations of RGS9:G5b ought to be carried out to more clearly discern the structural/dynamical similarities/differences between RGS7:Gb5 and RGS9:G5b. Analysis of the dynamic networks within RGS9:G5b should also be carried out to establish whether the results obtained for RGS7 can be extended to other members of the R7 family.

As requested, we performed new MD simulations along with the dynamical network analysis on the RGS9-GB5 complex. These new data, which are described in a new paragraph and compared to the RGS7-GB5 MD and dynamical network analysis results, indicate that the potential allosteric pathway linking the DHEX/DEP domains and the GGL-linker/RGS domains through GB5 noted for RGS7 is also conserved in RGS9 and thus could likely be universal for all R7 family members.